# Interleukin-17A-Correlated Myocarditis in Patients with Psoriasis: Cardiac Recovery following Secukinumab Administration

**DOI:** 10.3390/jcm12124010

**Published:** 2023-06-12

**Authors:** Andrea Frustaci, Nicola Galea, Lorenzo Dominici, Romina Verardo, Maria Alfarano, Rossella Scialla, Antonio Giovanni Richetta

**Affiliations:** 1IRCCS San Raffaele, Via di Val Cannuta, 00166 Rome, Italy; 2Department of Clinical, Internal, Anesthesiologist and Cardiovascular Sciences, Sapienza University, 00185 Rome, Italy; nicola.galea@uniroma1.it (N.G.); lorenzo.dominici@uniroma1.it (L.D.); maria.alfarano@uniroma1.it (M.A.); antonio.richetta@uniroma1.it (A.G.R.); 3Cellular and Molecular Cardiology Lab, IRCCS L. Spallanzani, 00149 Rome, Italy; romina.verardo@inmi.it (R.V.); rossella.scialla@inmi.it (R.S.)

**Keywords:** myocarditis, psoriasis, dilated cardiomyopathy, IL-17A

## Abstract

(1) Background: Psoriasis (PS) is a common immune-mediated disease of the skin with possible extension to joints, aorta and eye. Myocardial inflammation has rarely been suggested. (2) Aims: Report of PS-related myocarditis. (3) Methods and Results: One hundred consecutive patients with PS were screened for cardiac involvement. Among them, five male patients (aged 56 ± 9.5 years) with a moderate–severe form of PS showed dilated cardiomyopathy (LVEF < 35%) with normal coronary arteries and valves. They underwent a left-ventricular endomyocardial biopsy for evaluation of myocardial substrate. Endomyocardial samples were processed for histology and immunohistochemistry, including myocardial expression of Toll-Like Receptor 4 (TLR4) and interleukin-17A (IL-17A), which play a major role in PS pathogenesis. Real-time PCRs were carried out for cardiotropic viruses, and Western blot analysis was conducted for myocardial expression of IL-17A. Patients’ sera were tested for anti-heart autoantibodies. Active lymphocytic myocarditis was revealed in all five patients, characterized by an absence of viral genomes with PCR, positive anti-heart autoantibodies, overexpression of TLR-4 and enhancement of IL-17-A during western blot analysis, showing a 2.48-fold increase in psoriatic myocarditis compared with no psoriatic myocarditis and a six-fold increase compared to myocardial controls. Treatment included combination of prednisone (1 mg/kg daily for 4 weeks, tapered to 0.33 mg/kg) and azathioprine (2 mg/kg, daily) in 3 pts or secukinumab (SK, 150 mg/weekly for 4 weeks followed by 150 mg/monthly) in 2 pts for 6 months. LVEDD and LVEF improved in the first 3 pts (−14% and + 118%, respectively), while they completely recovered (LVEF > 50%) in the last 2 pts on SK. (4) Conclusions: IL-17A-related myocarditis can occur in up to 5% of patients with PS. It manifests as progressive dilated cardiomyopathy. It may completely recover following SK administration.

## 1. Introduction

Psoriasis (PS) is a common immune-mediated disease which affects 1–3% of the population [1]. The pathogenesis of PS is unknown. It is probably correlated to a complex interaction between environmental factors, genetic predisposition and characteristics of the immune system. Myocardial involvement in PS is poorly investigated. Single case reports and small observational studies suggest a possible relationship between PS and cardiomyopathy [2,3,4,5,6,7,8]. Eliakim-Raz et al. [2] reported the presence of cardiomyopathy in 20 patients (0.87%) among 2292 hospitalized patients with PS, 10 (0.43%) of which had dilated cardiomyopathy. However, the pathogenesis of this correlation is still unclear. IL-17A is the major cytokine implicated in PS pathogenesis, and it is overexpressed in psoriatic skin [1,9]. Secukinumab, an IL-17A-neutralizing IL17 antibody, has been demonstrated to have remarkable clinical efficacy in patients with moderate-to-severe PS [10,11].

The aim of the present study is to determine the incidence and pathogenesis of myocardial inflammation in patients with PS as well as its most appropriate treatment.

## 2. Materials and Methods

### 2.1. Patient Population

One hundred consecutive patients (62 M/38 F, mean age 55 ± 13.7 years) with a diagnosis of PS of various severities were screened for possible cardiac involvement. Severity of the disease was defined on the basis of PASI (Psoriasis Area Severity Index) [12,13] and on the presence of comorbidity. Among them, five male patients (mean age 56 ± 9.5 years) with moderate/severe PS were symptomatic for dyspnea and presented cardiac dilatation with remarkable reduction of ejection fraction, normal valves and absence of additional clinical problems and previous history of cardiac disease. To investigate the cause of cardiac compromise, those patients underwent non-invasive cardiac investigations including echocardiography and cardiac magnetic resonance (CMR) and an invasive cardiac study including coronary angiography and left-ventricular endomyocardial biopsy.

### 2.2. Clinical Studies

Clinical assessment, resting electrocardiogram, Holter monitoring, 2D-echocardiography and cardiac magnetic resonance (CMR) were performed at baseline in all patients. Cardiac catheterization, angiography, and left- or biventricular endomyocardial biopsy were also performed, after patients had given informed consent. The New York Heart Association (NYHA) classification was used to assess functional capacity determined by means of a questionnaire. The study complies with the Declaration of Helsinki, the locally appointed ethics committee (opinion number 2016-003014-28 (FARM12JCXN)) approved the research protocol and informed consent was obtained from all subjects.

### 2.3. Cardiac Magnetic Resonance (CMR)

CMR exams were performed on a 1.5 T scanner (Magnetom Avanto; Siemens Medical Systems, Erlangen, Germany) using body- and phase-array coils. The CMR protocol included the following elements: (i) cine-balanced steady-state free precession (cine-bSSFP) images were acquired during breath holds in the short-axis (from the base to the apex, 10–12 slices), two-chamber and four-chamber planes (TR: 51.3 ms; TE: 1.21 ms; flip angle: 45°; slice thickness: 8 mm; matrix: 256 × 256; field of view: 340–400 mm; and voxel size: 2.0 × 1.3 × 8.0 mm); (ii) black blood T2-weighted breath-hold short-tau inversion-recovery (T2w STIR) images were obtained using a segmented turbo spin-echo triple-inversion recovery technique in the short-axis (from the base to the cardiac apex, 8–10 slices), two-chamber and four-chamber planes (TR: 2 R–R intervals; TE: 75 ms; flip angle: 180°; TI: 170 ms; slice thickness: 8 mm; field of view: 340–400 mm; matrix: 256 × 256; and voxel size: 2.3 × 1.3 × 8 mm); (iii) an ECG-gated single-shot Modified Look-Locker inversion-recovery sequence with a 5(3)3 scheme and motion-correction postprocessing algorithm (Siemens WIP package no. 448) for native T1 mapping was acquired in short-axis basal, mid-ventricular and apical planes (TR: 314 ms; TE: 1.12 ms; flip angle: 35°; TI: 200 ms; slice thickness: 8 mm; field of view: 340–400 mm; matrix: 256 × 256; and voxel size: 2.1 × 1.4 × 8 mm); (iv) T2 mapping was acquired with a T2-3pt GRE in the short axis through basal, mid-ventricular and apical planes (TR: 239 ms; TE: 1.13 ms; flip angle: 12°; slice thickness: 8 mm; field of view: 340–400 mm; matrix: 256 × 256; and voxel size: 2.5 × 1.9 × 8 mm); and (v) late gadolinium-enhanced (LGE) imaging was performed 15 minutes after bolus injection of 0.1 mmol/kg of body weight of gadobutrol (Gadovist, Bayer AG, Leverkusen, Germany) at a flow rate of 2.0 ml/s, using a phase-sensitive inversion-recovery gradient-echo (PSIRGRE) sequence (TI: 250–300 ms; TR: 9.6 ms; TE: 4.4 ms; matrix: 256 × 208; flip angle: 25°; slice thickness: 8.0 mm; and slice spacing: 2.0 mm).

### 2.4. Endomyocardial Biopsy Studies

Endomyocardial biopsies (5–8 fragments/each patient) were performed in the septal–apical region of the left ventricle. Samples were either fixed in formalin and paraffin-embedded for histology and immunohistochemistry or snap frozen for molecular biology and western blot analysis.

### 2.5. Histology and Immunohistochemistry

For histological analysis, the endomyocardial samples were fixed in 10% buffered formalin and paraffin-embedded. Five-micron-thick sections were stained with haematoxylin and also eosin and Masson trichrome.

For the phenotypic characterization of the inflammatory infiltrates, immunohistochemistry for Cluster of Differentiation (CD)3, CD20, CD43, CD45RO and CD68 was performed (all Dako, Carpinteria, CA, USA). The presence of an inflammatory infiltrate of ≥14 leucocytes/mm^2^ including up to 4 monocytes/mm^2^, with the presence of CD3-positive T-lymphocytes of ≥7 cells/mm^2^ associated with evidence of degeneration and/or necrosis of the adjacent cardiomyocytes, was considered diagnostic for myocarditis. The number of CD3-positive cells was manually counted using a tally counter with a high power field (×400) scanning the entire slide. The area of tissue samples was measured by means of a computerized system (Imaging Software/NIS-Elements AR 4.30; Nikon Instruments Inc., Melville, NY, USA). The number of CD3-positive cells was expressed as the number of cells per square millimeter. Morphometric evaluation was performed by a pathologist who was blinded to the clinical data.

For the assessment of TLR4 and IL-17A, expression myocardial sections were incubated with a mouse monoclonal TLR4 antibody at 1:10 dilution (HTA125, Santa Cruz Biotechnology Inc., Dallas, TX, USA), a rabbit polyclonal IL 17/A antibody at 1: 250 dilution (Abcam Cambridge Biomedical Campus, Cambridge, UK) and a peroxidase/anti-peroxidase complex, followed by labelling with the chromogen diaminobenzidine (Dako, Agilent, Santa Clara, CA, USA). A semiquantitative evaluation of the cytoplasmic immunoreactivity for TLR4 and IL-17A (grading from 0 to 4) was applied. For TLR4 and IL-17A, grade 0 was attributed to absence of immunostaining, grade 1 to 1–10% positive cardiomyocytes, grade 2 to 11–30%, grade 3 to 31–60% and grade 4 to >60% positivity. Appropriate positive and negative controls were used, and tissues incubated with antibody diluent, without the primary antibody, were followed by incubation with secondary antibodies and detection reagents (negative control). In addition, samples were incubated with antibody diluent, supplemented with a non-immune immunoglobulin of the same isotype (IgG2a) and concentration as the primary monoclonal antibody. The samples were then incubated with the secondary antibody and detection reagents. These steps were adopted to rule out that what appears to be specific staining was not caused by non-specific interactions of immunoglobulin molecules with the samples.

### 2.6. Molecular Biology Studies

In all patients at baseline, a PCR for the most common cardiotropic viruses (adenovirus, enterovirus, influenza A and B viruses, Epstein–Barr virus, Parvovirus B19, Hepatitis C virus, Cytomegalovirus, Human Herpes Virus 6, and Herpes Simplex virus A and B) was performed to exclude myocardial viral infection. For the PCRs used in the clinic, quality controls were run regularly according to the hospital’s quality management system.

### 2.7. Western Blot Analysis

We determined the expression of IL-17A in frozen myocardial tissue. Results were compared with values from 10 surgical control patients with unloaded myocardium (papillary muscle of patients with mitral stenosis undergoing valve replacement) and 10 biopsy samples of patients with no psoriatic myocarditis.

### 2.8. Protein Isolation and Western Blot

Heart tissue samples were treated as previously described. The expression of Interleukin 17A, molecular weight 18 kDa, was visualized by using an anti-IL-17A antibody polyclonal (1:1000; Abcam Cambridge Biomedical Campus, UK). An anti-α-sarcomeric actin antibody (1:500; Sigma-Aldrich, St. Louis, MO, USA), molecular weight 43 kDa, was used for normalization. The signal was visualized using a secondary horseradish-peroxidase-labeled goat anti-rabbit antibody (goat anti-rabbit IgG-HRP 1:5000; Santa Cruz Biotechnology, Dallas, TX, USA) and enhanced chemiluminescence (ECL Clarity Bio-Rad, Munich, Germany). The purity as well as equal loading (40 γ) of the protein was determined by Nanodrop One (Thermofisher, Waltham, MA, USA). To normalize the target protein expression, the band intensity of each sample was determined by densitometry with Image J software (Version 1.53t 24 August 2022). Next, the intensity of the target protein was divided by the intensity of the loading control protein. This calculation adjusts the expression of the protein of interest to a common scale and reduces the impact of sample-to-sample variation. Relative target protein expression can then be compared across all lanes to assess changes in target protein expression across samples. Digital images of the resulting bands were quantified by the Image Lab software package (Image Lab Touch 3.0 Software, Bio-Rad Laboratories, Hercules, CA, USA) and expressed as arbitrary densitometric units.

### 2.9. Statistical Analysis

Statistical analysis was performed by the GraphPad Prism package, version 5.02 (Graphpad Software Inc., San Diego, CA, USA). Comparison between groups was performed with a Mann Whitney nonparametric test. A *p* value of less than 0.05 was considered statistically significant.

## 3. Results

### 3.1. Cardiac Studies

Among our psoriatic population (62 M/38 F, mean age 55 ± 13.7 years), 28% of patients showed mild skin involvement, while 56% had a moderate and 16% had a severe form of the disease; 23 patients also had arthritis, and 4 patients had ocular disease. Five male patients (mean age 56 ± 9.5 years) showed dilated cardiomyopathy (LVEF < 35%) with normal coronary arteries and valves. Their baseline clinical and instrumental data are reported in Table 1.

All five patients presented moderate/severe skin involvement. Two patients also had arthritis, and two patients had uveitis. The baseline NYHA Class was III/IV. Electrocardiograms documented premature ventricular contractions, supraventricular tachycardia and non-sustained ventricular tachycardia in three patients, atrial fibrillation in one patient and ST segment/T wave abnormalities in one patient (Figure 1A). At 2D-echo, all patients exhibited left-ventricular dilation (left-ventricular end-diastolic diameter, LVEDD = 64.6 ± 8.01 mm) with severe reduction of ejection fraction (LVEF = 20.2 ± 5.6%) and high filling pressure (E/E’ > 12 mmHg); moreover, right-ventricular dysfunction (tricuspid annular-plane systolic excursion, TAPSE < 16 mm) was present in three patients, and pericardial effusion was present in two patients. STIR T2-weighted CMR images demonstrated hyperintensity, and native T1 and T2 maps showed a diffuse increase in myocardial T1 and T2 values (Figure 1B,C) due to diffuse edema. CMR (Figure 1D,E and Appendix A) confirmed hypokinetic dilated cardiomyopathy. Late gadolinium-enhanced images did not show any fibrotic area (Figure 1F,G) except in two patients.

All five patients underwent a left-ventricular endomyocardial biopsy for evaluation of myocardial substrate. At cardiac catheterization, elevated filling pressures were documented (left-ventricular end-diastolic pressure, LVEDP > 12 mmHg) with normal coronary network at angiography. Endomyocardial biopsy was a safe procedure with no side effects in any patient.

### 3.2. Histology and Immunohistochemistry

All 5 pts exhibited virus-negative lymphocytic myocarditis with overexpression of TLR4 and IL-17A and positive anti-heart autoantibodies cross-reactive with skeletal muscle (Figure 2A–D and Figure 3A–D) [14].

Semiquantitative evaluation of immunostaining (grades from 0 to 4) for IL-17A and TLR4 showed an increased cardiomyocyte expression of IL-17A in PS myocarditis compared with no-psoriatic patients and vs. controls (respectively, 3.95 ± 0.13 vs. 1.91 ± 0.5 and 3.95 ± 0.13 vs. 0.19 ± 0.2) (Figure 3E).

Immunostaining for TLR4 showed an overexpression of TLR4 in PS patients vs. no-PS myocarditis and vs. controls (3.44 ± 0.51 vs. 1.56 ± 0.2 and 3.44 ± 0.51 vs. 0.18 ± 0.17).

### 3.3. Western Blot

Protein analysis showed a 2.48-fold increase in IL-17A protein expression in PS myocarditis compared with no-PS myocarditis and a six-fold increase when compared with controls (respectively, 51,560 ± 899 vs. 20,734.5 ± 2502, *p* = 0.037, *p* < 0.01 and 51,560 ± 899 vs. 8600 ± 1055, *p* = 0.005, *p* < 0.001).

### 3.4. Treatment

All five patients showed virus-negative lymphocytic myocarditis; three were treated for 6 months with immunosuppressive therapy including 1 mg/kg prednisone daily for 4 weeks followed by 0.33 mg/kg daily for an additional 5 months and 2 mg/kg azathioprine daily for 6 months according to TIMIC (Tailored Immunosuppression in virus-negative Inflammatory Cardiomyopathy) protocol [15], and two patients were treated with secukinumab, an IL-17A inhibitor (150 mg/weekly for 4 weeks followed by 150 mg/monthly for 5 months). All patients also received appropriate therapy for heart failure with beta-blockers, angiotensin receptor–neprilysin inhibitors/ACE inhibitors/angiotensin receptor blockers, mineralocorticoid receptor antagonists and diuretics according to current ESC Guidelines [16].

### 3.5. Follow-Up

At 6-month follow-up (Table 2 and Figure 4), three patients treated with prednisone and azathioprine improved with NYHA Class reduction from III to II and LVEF increased from 22.6% to 48.6%, while two patients on SK recovered completely with a reduction of NYHA Class from III/IV to I, LVEED decreasing from 66 to 53 mm and normalization of LVEF from 16% to 55% (Figure 1H, I and Appendix A). Arrhythmias and ST segment/T-wave abnormalities (Figure 1J) on their electrocardiograms disappeared. CMR showed resolution of edema on a STIR T2-weighted image and reduction of native T1 and T2 values on mapping sequences. Patients 1 and 5 exhibited persistence of late gadolinium enhancement as a consequence of myocardial fibrosis. Skin lesions almost completely disappeared (Figure 2B), while arthritis improved and eyes were healed.

CineMR dynamic images of patient 5 were acquired at baseline during a subacute phase of psoriasis-related myocardial injury (upper row) and at 18-months follow-up (lower row), after IL-17A monoclonal antibody therapy. Images acquired in the mid-ventricular short-axis and four-chamber views show a marked reduction of LV function and LV dilation at baseline (EF: 16%), with total recovery of contractile function at follow-up (EF: 55%).

## 4. Discussion

PS is an immune-mediated inflammatory disease with prevalent skin involvement. Indeed, a systemic spreading of the disease has been recognized with possible extension to joints [17], aorta [18] and eye [19]. Myocardial inflammation has been previously suggested in the setting of dilated cardiomyopathy manifesting in subjects with PS, but histologic confirmation as well as pathogenetic correlations with the former disease have not been provided so far.

This is the first study that analyzes the incidence of cardiomyopathy in the context of a large population affected by PS, its histologic and molecular substrate obtained by means of endomyocardial biopsy and finally its most appropriate treatment.

In the present report, cardiomyopathy has been recognized in 5 of 100 consecutive patients with PS. These five male patients had a moderate-to-severe form of the disease with involvement of skin and other districts: joints in two patients and eyes in two patients. Histology of endomyocardial samples documented virus-negative lymphocytic myocarditis with overexpression of IL-17A. These findings were similar to those observed in the corresponding skin biopsy (Figure 2A) and suggest myocardial inflammation to be an extension of PS to the heart. Overexpression of IL-17A was ascertained by both immunohistochemistry and western blot analysis, which showed a 2.5-fold increase in the value obtained in no-psoriatic myocarditis and a six-fold increase in the normal myocardial controls. Immuno-pathogenesis of myocardial inflammation has also been supported by myocardial positivity of TLR4 at immunohistochemistry as well as the presence in the patients’ sera of anti-heart abs which cross-reacted with skeletal muscle (Figure 3D). This last finding suggests that symptoms connected with skeletal muscle damage such as pain, cramps and functional impairment may accompany the manifestations of severe forms of PS. The positivity of anti-heart abs suggests an autoimmune mechanism in myocardial inflammation responsive to immunosuppressive therapy [14], also supported by the cross-reactivity with skeletal muscle. This could correlate with cytokine activation, in particular with IL-17A, which is the main cytokine involved in PS.

As far as treatment of PS-related inflammatory cardiomyopathy is concerned, although limited to few cases, two alternatives have been tested including the TIMIC protocol [15] (combination of prednisone and azathioprine) commonly adopted for virus-negative inflammatory cardiomyopathy, and secukinumab, a recombinant, high-affinity, fully human immunoglobulin G1κ monoclonal antibody that selectively binds and neutralizes interleukin-17A [10,11], which has controlled skin lesions in advanced PS. In this report, although both approaches have given positive results, administration of secukinumab for 6 months has been followed by resolution of skin lesions (Figure 2B) and normalization of cardiac parameters with LVEF rising from 16 up to 55%. No side effects were reported from the two types of treatment.

Practical Implications

Immune-mediated myocardial inflammation can be part of systemic manifestations that apply to the advanced forms of PS. While suspicion of its presence can arise following application of cardiac magnetic resonance, confirmed recognition requires an endomyocardial biopsy showing virus-negative inflammatory cardiomyopathy characterized by overexpression of IL-17A. Although immunosuppression is clinically beneficial, treatment with secukinumab appears to be the most effective therapeutic solution with complete recovery of skin and cardiac damage.

## 5. Conclusions

IL-17A-related myocarditis can occur in up to 5% of patients with PS. It manifests as progressive dilated cardiomyopathy. It may completely recover following SK administration.

## Figures and Tables

**Figure 1 jcm-12-04010-f001:**
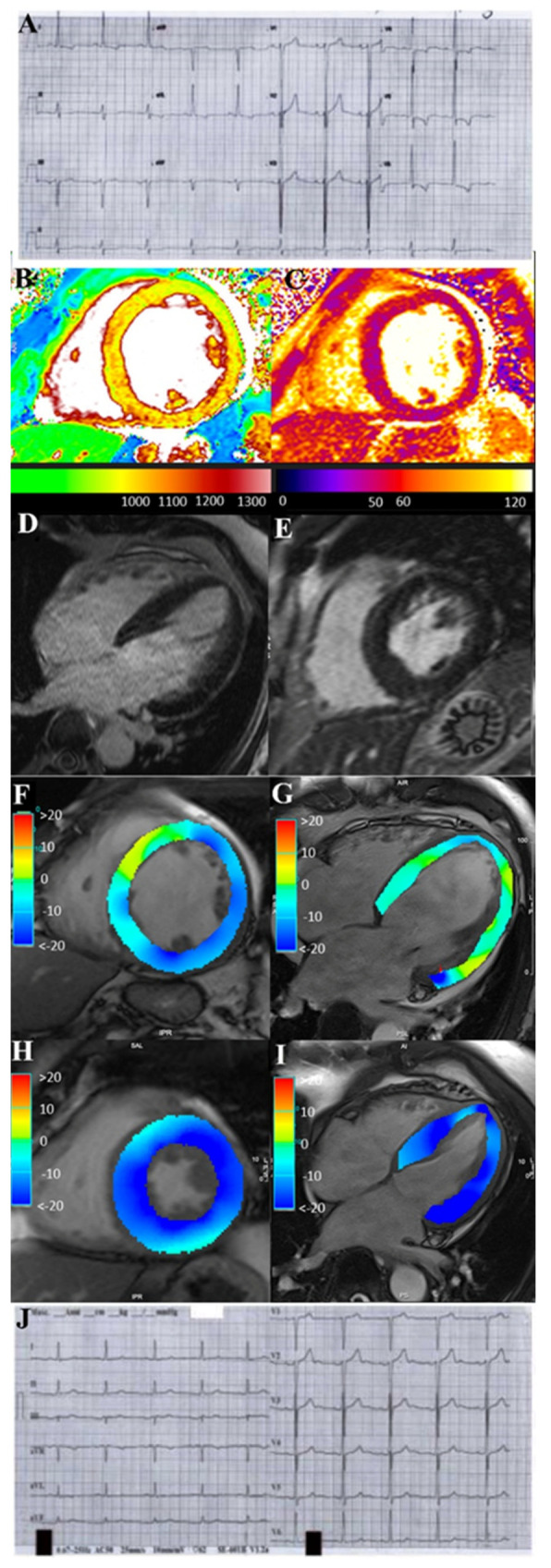
ECG (panels (**A**,**J**)) and CMR (panels (**B**−**I**)) findings of pt 5 with severe PS−related myocarditis at baseline and following 6 months’ SK administration. ECG shows complete normalization of ST−T abnormalities after therapy (panel (**A**): ECG before therapy and panel (**J**): ECG post therapy). CMR: native T1 (**B**) and T2 (**C**) maps show a diffuse increase in myocardial T1 and T2 values due to diffuse edema during acute phase (**B**,**C**). CineMR images, acquired in mid-ventricular short-axis (**D**) and four−chamber (**E**) views at end−systolic phase show a marked reduction in LV function, respectively visualized as impairment of GLS and GCS strain values. Late gadolinium-enhanced images (**F**,**G**) do not show any fibrotic or necrotic myocardial areas. CineMR images at follow−up (**H**,**I**) show a complete recovery of LV function with normal GCS and GLS strain values. LV: left ventricle; GCS: global circumferential strain; GLS: global longitudinal strain.

**Figure 2 jcm-12-04010-f002:**
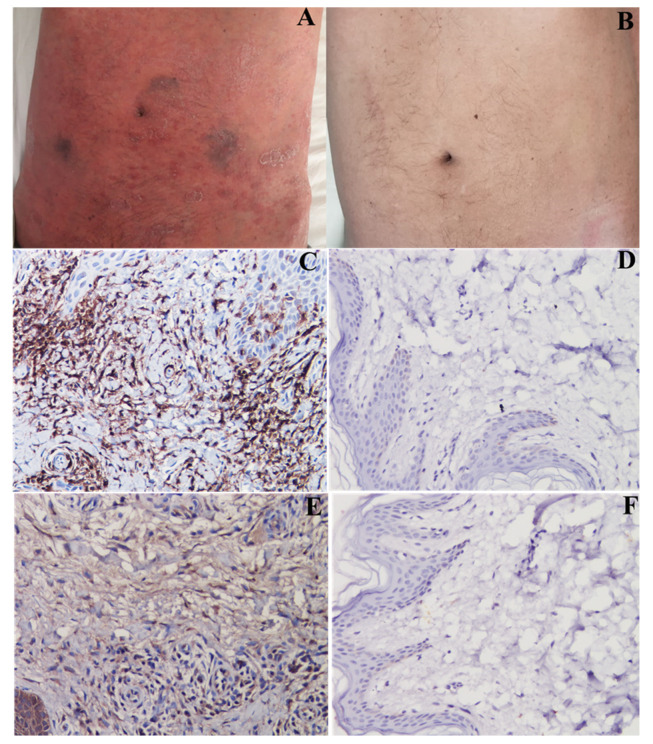
Severe psoriatic skin lesions from pt 5 that completely recover after SK administration. Panels (**A**–**B**): skin dermatitis (**A**,**B**) and its recovery. Panels (**C**–**F**): complete recovery from skin inflammation. ((**C**): extensive T-lymphocytic (CD45Ro+) inflammatory infiltrates); (**E**): overexpression of IL-17A that disappears (panels (**D**,**F**), respectively) after treatment.

**Figure 3 jcm-12-04010-f003:**
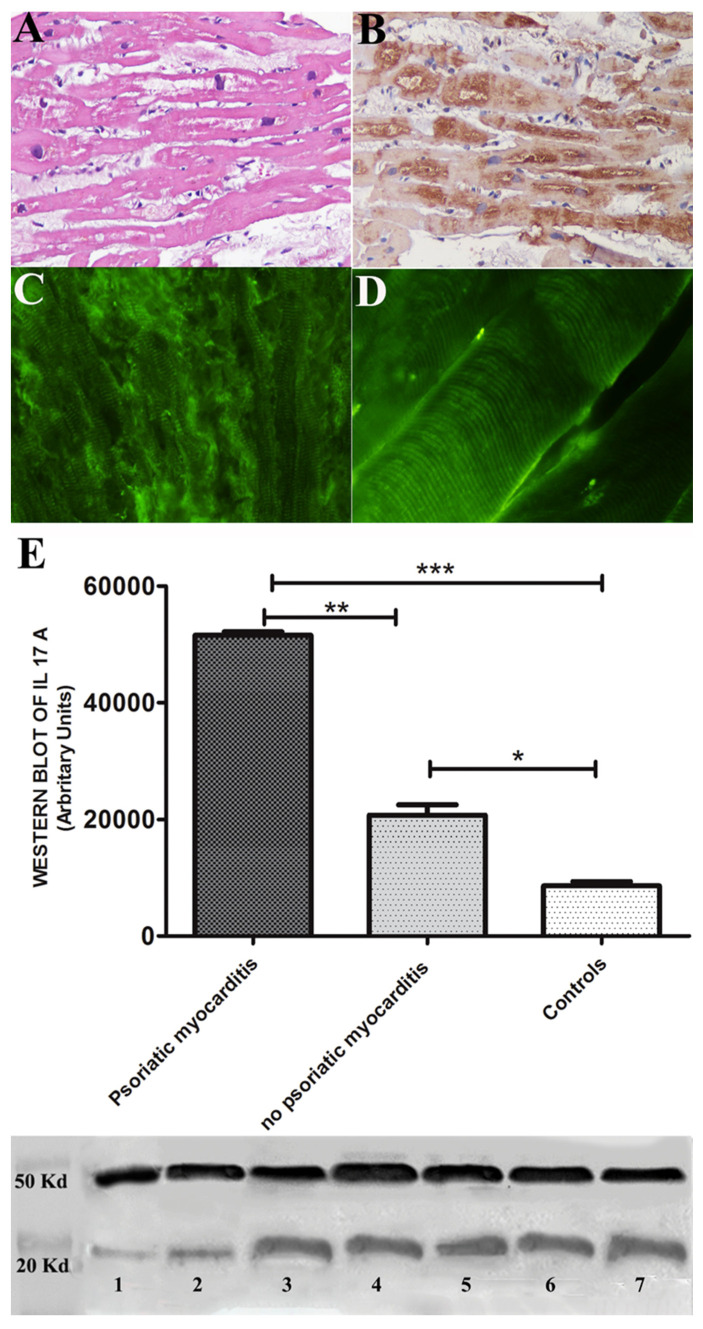
Endomyocardial biopsy findings from pt 5 with PS-related myocarditis including histology (panel (**A**)). Immunohistochemistry for IL-17A (panel (**B**)) and anti-heart abs (panels (**C**,**D**)). It shows active lymphocytic myocarditis with overexpression of IL-17A and positive anti-heart abs (panel (**C**)), cross-reacting with skeletal muscle (panel (**D**)). Graphs document western blot of IL-17A (18 Kda) and show a 2.48-fold increase in IL-17A protein expression in psoriatic myocarditis compared with no psoriatic myocarditis and a six-fold increase when compared with controls (respectively, 51,560 ± 899 vs. 20,734.5 ± 2502, *p* = 0.037, *p* < 0.01 and 51,560 ± 899 vs. 8600 ± 1055, *p* = 0.005, *p* < 0.001). Alpha sarcomeric actin (43 kDa) was used as a loading control (panel (**E**)). Marker: lane 1 = normal control, lane 2 = no psoriatic myocarditis, lane 3 = pt 1, lane 4 = pt 2, lane 5 = pt 3, lane 6 = pt 4 and lane 7 = pt 5. * *p* < 0.05; ** *p* < 0.01; *** *p* < 0.001.

**Figure 4 jcm-12-04010-f004:**
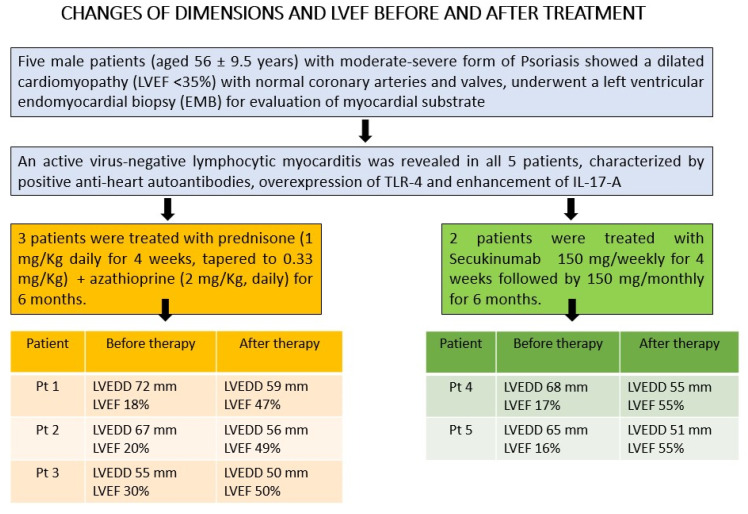
Changes in dimensions and LVEF at 2D-echocardiogram before and after treatment. LVEDD, left-ventricular end-diastolic diameter; LVEF, left-ventricular ejection fraction.

**Table 1 jcm-12-04010-t001:** Baseline clinical and instrumental data of the five patients with psoriasis-related myocarditis.

	Patient 1	Patient 2	Patient 3	Patient 4	Patient 5
**Age/sex**	67/M	52/M	54/M	65/M	44/M
**Severity of skin psoriasis**	Moderate	Severe	Moderate	Moderate	Severe
**Presence of arthritis**	No	No	Yes	No	Yes
**Eye involvement**	No	No	No	Yes, uveitis	Yes, uveitis
**NYHA Class**	III	III	III	III	IV
**ECG**	PVC	AF	SVT and NSVT	NSVT	ST-T abn
**2D-echocardiogram**					
***LVEDD* (mm)**	72	67	55	68	65
***LVEF* (%)**	18	20	30	17	16
***Left atrium* (volume, mL)**	60	68	50	70	126
** *E/E’ ratio* **	30	28	22	25	29
***TAPSE* (mm)**	14	13	24	18	16
** *Pericardial effusion* **	No	No	No	Yes	Yes
**Cardiac magnetic resonance**					
***LVEDV* (mL/m^2^)**	108.7	131.0	130.1	114.08	217.5
***LVESV* (mL/m^2^)**	91.5	110.6	108.2	95.6	181.0
***LVEF* %**	15.8	20.3	32.7	16.1	16.7
** *MWT* **	13	11	10	11	13
***LV mass* (g/m^2^)**	79	85.6	97.1	80.2	158.1
** *Edema* **	No	No	No	Yes	Yes
** *LGE* **	Yes	No	No	Yes	Yes
***T1 global ± SD* (ms)**	1285 ± 131	1015 ± 153	1020 ± 81	1056 ± 96	1071 ± 52
***T2 global ± SD* (ms)**	56 ± 14	52 ± 9	51 ± 6	50 ± 8	50 ± 6
**Coronary arteries**	Normal	Normal	Normal	Normal	Normal

PVC, premature ventricular contractions; AF, atrial fibrillation; SVT, supraventricular tachycardia; NSVT, non-sustained ventricular tachycardia; ST-T abn, ST segment/T-wave abnormalities; LVEDD, left-ventricular and diastolic diameter; LVEF left-ventricular ejection fraction; E/E’ ratio, ratio of mitral peak velocity of early filling (E) to early diastolic mitral annular velocity (E’); TAPSE, tricuspid annular-plane systolic excursion; LVEDV, left-ventricular end-diastolic volume; LVESV, left-ventricular end-systolic volume; LVM, left-ventricular mass; LGE, late gadolinium enhancement; SD, standard deviation.

**Table 2 jcm-12-04010-t002:** Six-month follow-up clinical and instrumental data of the five patients with psoriasis-related myocarditis.

	Patient 1	Patient 2	Patient 3	Patient 4	Patient 5
**Age/sex**	67/M	52/M	54/M	65/M	44/M
**Therapy**	Az + Pr	Az + Pr	Az + Pr	SK	SK
**Severity of skin psoriasis**	Mild	Mild	Mild	Healed	Healed
**Presence of arthritis**	No	No	Yes (improved)	No	Yes (improved)
**Eye involvement**	No	No	No	Healed	Healed
**NYHA Class**	II	II	II	I	I
**ECG**	Normal	Sinus rhythm	SVT, PVC	Normal	Normal
**2D-Echocardiogram**					
***LVEDD* (mm)**	59	56	50	55	51
***LVEF* (%)**	47	49	50	55	55
***Left atrium* (volume, mL)**	62	45	53	40	50
** *E/E’ ratio* **	13	11	12	10	8
***TAPSE* (mm)**	19	20	23	25	24
** *Pericardial effusion* **	No	No	No	No	No
**Cardiac magnetic resonance**					
***LVEDV* (mL/m^2^)**	98.7	83.0	72.5	99.9	82.7
***LVESV* (mL/m^2^)**	52.6	42.0	34.4	45.2	37.9
***LVEF* %**	46.7	49.2	50.5	54.3	55.2
** *MWT* **	14	11	9	11	13
***LV mass* (g/m2)**	75	68.5	63	69.3	88.5
** *Edema* **	No	No	No	No	No
** *LGE* **	Yes	No	No	Yes	Yes
***T1 global ± SD* (ms)**	1010 ± 117	970 ± 101	982 ± 106	980 ± 102	1030 ± 62
***T2 global ± SD* (ms)**	41 ± 5	46 ± 6	49 ± 9	42 ± 5	46 ± 7

Az, azathioprine; Pr, prednisone; SK, secukinumab; SVT, supraventricular tachycardia; PVC, premature ventricular contraction; LVEDD, left-ventricular and diastolic diameter; LVEF, left-ventricular ejection fraction; E/E’ ratio, ratio of mitral peak velocity of early filling (E) to early diastolic mitral annular velocity (E’); TAPSE, tricuspid annular-plane systolic excursion; LVEDV, left-ventricular end-diastolic volume; LVESV, left-ventricular end-systolic volume; MWT, maximal wall thickness; LV mass, left-ventricular mass; LGE, late gadolinium enhancement; SD, standard deviation.

## Data Availability

The datasets used and analyzed during the current study are available from the corresponding author upon reasonable request.

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
