# Peer review of "Interleukin-17A-Correlated Myocarditis in Patients with Psoriasis: Cardiac Recovery following Secukinumab Administration"

_jcm, 2023, doi:10.3390/jcm12124010_

Round 1
Reviewer 1 Report
The authors present a very exciting study of cardiac involvement in psoriasis/psoriatic arthritis. 5 patients were identified, which were analyzed in detail including biopsy of the heart. Even though it is only a small group, this study is certainly very valuable because of the invasive diagnostics. The therapeutic success shown with an IL-17A inhibitor is groundbreaking. Wonderful study.
I have the following points to be clarified:
Did all 5 patients not know about their cardiac disease before? Were they asymptomatic?
No-Psoriatic myocarditis and controls like shown in fig 2E, where are these patients described? How many, characteristics?
Table 1: dry eye as „eye involvement“ in Psoriasis/psoriatic arthritis is not a usual symptom that clearly belongs to psoriasis/psoriatic arthritis. Normally one thinks of uveitis in the case of eye involvement. Please explain what is meant, and there was no uveitis? You cite one paper suggesting an association between dry eyes and Psoriasis, but dry eyes are a very common symptom. I suggest dry eyes not be labelled as eye involvement, if it is to be included then it can be labelled directly in the table as dry eyes and not eye involvement. And then please add the presence of uveitis yes/no.
Page 6 line 255: “IL-17A inhibitor (100 mg monthly for 6 months)”: please specify which IL-17A inhibitor was used (different substances available and 100mg monthly is an unusual dosage for all of them, might be different in different countries). In the discussion section you write it was secukinumab, but this is applicated with 150mg monthly after a saturation period of 150mg/week for 5 weeks.
Please expand the discussion to anti-heart antibodies. Where do you usually find them? What are they doing in the pathogenesis in other patients? How do you envision the development of these antibodies in IL-17 driven disease?
The changes bevor and after treatment are summarized in Table 1 and 2. I would suggest to visualize this partly in a figure demonstration the changes for special parameters, like for example for the impressive improvement in LVEF% (before/after for all 5 patients for example in different colours dependent on treatment)
Minor points:
Page 1, line 24-24: „Among them, five male patients (aged 56 ± 9.5 years) with moderate- severe form showed a dilated cardiomyopathy…. „ please specify „a moderate-severe form of PS, otherwise it could be also meant a moderate-severe form of cardiac involvement
Page 4, line 146-149, please specify the PCR, you do not have to show the primers and the PCR protocol, but a statement to ensure the quality like „PCR used in the clinic, quality controls run regularly according to the hospital's quality management system“ would be nice.
English is legible overall.
Author Response
The authors present a very exciting study of cardiac involvement in psoriasis/psoriatic arthritis. 5 patients were identified, which were analysed in detail including biopsy of the heart. Even though it is only a small group, this study is certainly very valuable because of the invasive diagnostics. The therapeutic success shown with an IL-17A inhibitor is ground-breaking. Wonderful study.
Reply: We sincerely appreciate the reviewer’s considerations.
I have the following points to be clarified:
Did all 5 patients not know about their cardiac disease before? Were they asymptomatic?
Reply: all 5 patients were symptomatic for dyspnea; no patient had had a prior history of heart problems. It has been added in the text.
No-Psoriatic myocarditis and controls like shown in fig 2E, where are these patients described? How many, characteristics?
Reply: the expression of myocardial IL17A was compared with biopsy sample from patients with no-psoriatic myocarditis and controls. It has been added the sentence in the method regarding western blot: “We determined the expression of IL-17A in frozen myocardial tissue. Results were compared with values from surgical 10 control unloaded myocardium (papillary muscle of patients with mitral stenosis undergoing valve replacement) and 10 biopsy samples of patients with no psoriatic myocarditis”.
Table 1: dry eye as „eye involvement“ in Psoriasis/psoriatic arthritis is not a usual symptom that clearly belongs to psoriasis/psoriatic arthritis. Normally one thinks of uveitis in the case of eye involvement. Please explain what is meant, and there was no uveitis? You cite one paper suggesting an association between dry eyes and Psoriasis, but dry eyes are a very common symptom. I suggest dry eyes not be labelled as eye involvement, if it is to be included then it can be labelled directly in the table as dry eyes and not eye involvement. And then please add the presence of uveitis yes/no.
Reply: eye involvement was caused by uveitis, it has been specified in the text and in the table.
Page 6 line 255: “IL-17A inhibitor (100 mg monthly for 6 months)”: please specify which IL-17A inhibitor was used (different substances available and 100mg monthly is an unusual dosage for all of them, might be different in different countries). In the discussion section you write it was secukinumab, but this is applicated with 150mg monthly after a saturation period of 150mg/week for 5 weeks.
Reply: I apologize for the incorrect indication; Secukinumab was applicated 150 mg/week for 4 weeks followed by 150 mg/monthly for 5 months (for a total of 6 months of therapy). It has been added in the text.
Please expand the discussion to anti-heart antibodies. Where do you usually find them? What are they doing in the pathogenesis in other patients? How do you envision the development of these antibodies in IL-17 driven disease?
Reply: the discussion has been expanded. A reference has been added (Caforio ALP, Adler Y, Agostini C, Allanore, et al. Diagnosis and management of myocardial involvement in systemic immune‐mediated diseases: a position statement of the European Society of Cardiology Working Group on Myocardial and Pericardial Disease. Eur Heart J. 2017; 38: 2649–2662).
The changes before and after treatment are summarized in Table 1 and 2. I would suggest to visualize this partly in a figure demonstration the changes for special parameters, like for example for the impressive improvement in LVEF% (before/after for all 5 patients for example in different colours dependent on treatment)
Reply: The figure demonstration (figure 4) has been added in the text
Minor points:
Page 1, line 24-24: „Among them, five male patients (aged 56 ± 9.5 years) with moderate- severe form showed a dilated cardiomyopathy…. „ please specify „a moderate-severe form of PS, otherwise it could be also meant a moderate-severe form of cardiac involvement
Reply: the sentence has been modified as follows: “Among them, five male patients (aged 56 ± 9.5 years) with moderate-severe form of PS showed a dilated cardiomyopathy (LVEF <35%) with normal coronary arteries and valves”
Page 4, line 146-149, please specify the PCR, you do not have to show the primers and the PCR protocol, but a statement to ensure the quality like „PCR used in the clinic, quality controls run regularly according to the hospital's quality management system“ would be nice.
Reply: the sentence has been added in the text
Reviewer 2 Report
Frustaci and colleagues provided a case-serie of 5 patients presenting with psoriasis complicated by myocarditis, treated with Secukinumab. The author emphasized the role of interleukine 17A in the pathogenesis of the psoriasis and the benefit of acting on this pathway in order to control cardiac-related symptom of the disease. The article is overall well-written. The figures and images are really educational. I only have a few comments.
- Introduction section. Please use the abbreviation “PS” further in the text (e.g. line 58).
- Page 2, lines 64-66: “In particular 28% of patients showed mild skin involvement, while 56% had a moderate and 16% a severe form of the disease; 23 patients had also arthritis and 4 patients ocular disease.” Please put this sentence in the results section of the manuscript.
- Page 5, line 185. The authors state in the text that one patient presented eye disease, but 2 patients are mentioned in Table 1.
- Page 5, lines 191-192. The authors state in the text that right ventricular dysfunction (tricuspid annular plane systolic excursion, TAPSE < 15 mm) was present in 3 patients but only 2 patients are mentioned in Table 1.
- Page 6, line 224. Can the authors define the abbreviation of TIMIC?
- The video 1 was unfortunately not available. Could you please provide it with the revised version?
- Could the authors add the physiological mechanism of the Secukinumab molecule to the discussion section of the manuscript?
- Page 11, legend of Table 1. I would not use the additional symbols to provide a legend but just the same abbreviations than in the Table, to make it easier for the reader. Same comment for page 13, legend of Table 2.
Minor editing of English language required.
Author Response
Frustaci and colleagues provided a case-serie of 5 patients presenting with psoriasis complicated by myocarditis, treated with Secukinumab. The author emphasized the role of interleukine 17A in the pathogenesis of the psoriasis and the benefit of acting on this pathway in order to control cardiac-related symptom of the disease. The article is overall well-written. The figures and images are really educational. I only have a few comments.
- Introduction section. Please use the abbreviation “PS” further in the text (e.g. line 58).
Reply: It has been done as requested
- Page 2, lines 64-66: “In particular 28% of patients showed mild skin involvement, while 56% had a moderate and 16% a severe form of the disease; 23 patients had also arthritis and 4 patients ocular disease.” Please put this sentence in the results section of the manuscript.
Reply: It has been done as requested
- Page 5, line 185. The authors state in the text that one patient presented eye disease, but 2 patients are mentioned in Table 1.
Reply: We apologize for the mistake. The patients were two and had uveitis.
- Page 5, lines 191-192. The authors state in the text that right ventricular dysfunction (tricuspid annular plane systolic excursion, TAPSE < 15 mm) was present in 3 patients but only 2 patients are mentioned in Table 1.
Reply: We apologize for the mistake. The patients were 3 because we considered right ventricular dysfunction in case of tricuspid annular plane systolic excursion (TAPSE) < 16 mm and not 15 mm. The value has been corrected in the text
- Page 6, line 224. Can the authors define the abbreviation of TIMIC?
Reply: the abbreviation of TIMIC has been clarified in the text: Tailored IMmunosuppression in virus-negative Inflammatory Cardiomyopathy
- The video 1 was unfortunately not available. Could you please provide it with the revised version?
Reply: The video 1 is now provide
- Could the authors add the physiological mechanism of the Secukinumab molecule to the discussion section of the manuscript?
Reply: it has been done as requested
- Page 11, legend of Table 1. I would not use the additional symbols to provide a legend but just the same abbreviations than in the Table, to make it easier for the reader. Same comment for page 13, legend of Table 2.
Reply: the legend of table 1 and table 2 has been modified as requested
Round 2
Reviewer 1 Report
Thank you, all points were clarified.